# RBF-PINN: NON-FOURIER POSITIONAL EMBEDDING IN PHYSICS-INFORMED NEURAL NETWORKS

**Chengxi Zeng,**[*] **Tilo Burghardt & Alberto M. Gambaruto**
University of Bristol, UK
{cz15306, tb2935, alberto.gambaruto}@bristol.ac.uk

## ABSTRACT

While many recent Physics-Informed Neural Networks (PINNs) variants have had considerable success in solving Partial Differential Equations, the empirical benefits of feature mapping drawn from the broader Neural Representations research have been largely overlooked. We highlight the limitations of widely used Fourier-based feature mapping in certain situations and suggest the use of the conditionally positive definite Radial Basis Function. The empirical findings demonstrate the effectiveness of our approach across a variety of forward and inverse problem cases. Our method can be seamlessly integrated into coordinate-based input neural networks and contribute to the wider field of PINNs research.

## 1 INTRODUCTION

Many scientific phenomena can be described by sets of differential equations (DEs). The prior physics knowledge is then formulated as regularisers that can be embedded in modern machine learning algorithms supported by the Universal Approximation Theorem Hornik et al. (1989). Physics-Informed Machine Learning (PIML) Karniadakis et al. (2021) is a learning paradigm that combines data-driven models with physical laws and domain knowledge to solve complex problems in science and engineering. It has recently attained remarkable achievements in a wide range of scientific research such as Electronics Smith et al. (2022); Hu et al. (2023); Nicoli et al. (2023), Dynamical System Thangamuthu et al. (2022); Ni & Qureshi (2023), Meteorology Kashinath et al. (2021); Giladi et al. (2021) and Medical Image Goyeneche et al. (2023); Salehi & Giannacopoulos (2022); Pokkunuru et al. (2023). One of the leading methods is called Physics-Informed Neural Networks (PINNs) Raissi et al. (2019). By adding the DEs as penalty terms in the deep Neural Networks (NN), it exploits the differentiability of NN to compute derivatives of the explicit functions and introduces domain-specific regularisation during optimisation. Adhering to conventional solvers, the PINNs formulation necessitates the specification of initial/boundary conditions (IC/BC) within a confined spatial-temporal domain. Boundary sampling points and domain collocation points are used to evaluate the residuals of the conditions and DEs via an overparameterised NN. The objective is to optimise the NN by minimising the residuals, resulting in a converged parameter space. This parameter space can then serve as a surrogate model that accurately represents the solution space of the DEs. The PINNs can be formulated as follows:

$$\mathcal{D}[u(x,t;\alpha_i)] = F(x,t), t \in \mathcal{T}[0,T], \forall x \in \Omega \text{ and } \mathcal{B}[u(x,t)] = H(x,t), t \in \mathcal{T}[0,T], x \in \partial\Omega. \quad (1)$$

where $\mathcal{D}[\cdot]$ is the differential operator and $\mathcal{B}[\cdot]$ is the boundary operator, $x$ and $t$ are the independent variables in spatial and temporal domains $\Omega$ and $\mathcal{T}$, respectively. The $\alpha_i$ are coefficients of the DE system and remain wholly or partially unknown in Inverse Problems. For time-dependent PDEs, the initial condition can be treated as a special type of BC. $F$ and $H$ are arbitrary functions.

PINNs are parameterised by $\theta$, the solution space represented can give numerical solution $\hat{u}_\theta$ at any $x$ and $t$ within the domain. And the training loss functions are defined as follows:

$$\mathcal{L}(\theta; \mathcal{X}_{(x,t)}) = \frac{\lambda_r}{N_r} \sum_{i=1}^{N_r} \left| \mathcal{D}[\hat{u}_\theta(x_r^i)] - F(x_r^i) \right|^2 + \frac{\lambda_{bc}}{N_{bc}} \sum_{i=1}^{N_{bc}} \left| \mathcal{B}[\hat{u}_\theta(x_{bc}^i)] - H(x_{bc}^i) \right|^2 \quad (2)$$

where $\{x_r^i\}_{i=1}^{N_r}$ and $\{x_{bc}^i\}_{i=1}^{N_{bc}}$ are domain collocation points and boundary points, they are evaluated by computing the mean squared error. $\lambda_r$ and $\lambda_{bc}$ are the corresponding weights of each term.

---

[*]Correspondence Author. The full version of the paper can be found in arxiv.org/abs/2402.06955. The code can be found in repo github.com/SimonZeng7108/RBF-PINN/tree/master.

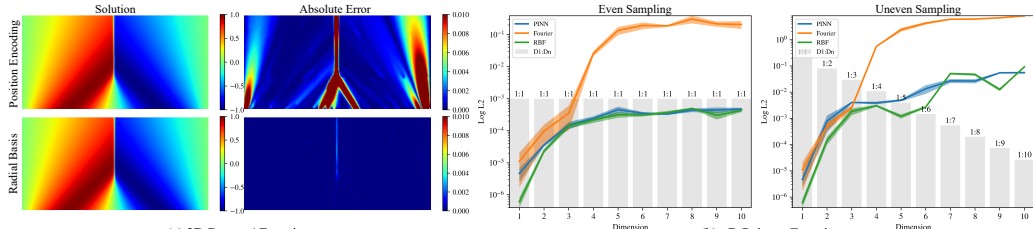

Figure 1: (a) Solutions using PINNs to solve Diffusion Equation with Positional Encoding (Top) and Our RBF (Bottom) feature mappings ; (b) $L2$ error on nD Poisson equation from 1 to 10 dimensions.

Although significant advances have been achieved in prior research, there has been limited exploration of feature mapping, with only a few studies Wang et al. (2021b); Wong et al. (2022) recognising its potential within PINNs. Originally introduced in Natural Language Processing (NLP), feature mapping aims to map the input to a higher-dimensional feature space for a better neural representation. It was subsequently identified as an effective approach for mitigating spectral bias in visual representations Tancik et al. (2020). Feature mapping is a broader term for positional encoding that can involve either fixed encoding or trainable embedding. This simple one-layer projection can map the spatio-temporal input $\mathbf{x}$ to a higher dimension feature space, $\Phi : \mathbf{x} \in \mathbb{R}^n \to \mathbb{R}^m$, and typically $n \ll m$.

In PINNs, Wang et al. (2020) has leveraged the 'Neural Tangent Kernel' (NTK) theory that reveals PINNs suffer from 'Spectral Bias' in the infinite-width limit. The NTK exhibits sensitivity to both input and network parameters. Its trait is particularly contingent on factors such as the input gradient of the model, the variance at each layer, and the nonlinear activations. Hence, the training dynamics of PINNs are significantly influenced by the input before parameterised layers.

Our contribution in this paper can be summarised as: First, we show the limitations and shortcomings of the widely used Fourier-based feature mappings in some Partial Differential Equations (PDEs) and thoroughly benchmark a wide range of feature mapping methods, including some which have not been employed in PINNs before. Secondly, we present a framework for designing feature mapping functions and introduce a conditional positive definite Radial Basis Function. This method surpasses Fourier-based feature mapping in various forward and inverse tasks.

## 2   LIMITATION OF FOURIER FEATURES

Basic Fourier features can lead to undesired artifacts, as illustrated in Figure 1(a) (detailed equations can be found in Appendix G). When applying Positional Encoding to address Burgers' equations, a notable prediction error is observed in the area approaching a jump solution. This phenomenon is akin to the Gibbs phenomenon in signal processing, where the approximated function value by a finite number of terms in its Fourier series tends to overshoot and oscillate around a discontinuity.

Another unexpected experimental result that shows poor performance is observed when utilising Random Fourier Features Tancik et al. (2020) in high-dimensional problems, as depicted in Figure 1(b). Two cases are set up, one case with each dimension of evenly sampled points and another with uneven sampling points, where the number of sampling points $x_r$ is set as $\frac{1}{D}$. The latter example resembles the unsteady Navier-Stokes equations for fluid dynamics, which have dense samples in the spatial domains, but sparse sampling in the temporal dimension. Despite hyperparameter tuning for the Fourier feature, including the arbitrary scale $\sigma$ and the number of Fourier features, none of the attempts reduce the elevated error in high-dimensional cases. The experiments are repeated with 3 random seeds, and standard deviations are displayed in the highlighted area.

## 3   PROPOSED METHOD

In NTK theory, the multi-layer perceptions (MLPs) function is approximated by the convolution of the stationary composed NTK function $K_{\text{COMP}} = K_{NTK} \circ K_\Phi$ with weighted Dirac delta over the input $\mathbf{x}$ (background in Spectral Bias and Composed NTK are in Appendix C), we can formulate the

$K_{\text{COMP}}$ by:

$$
\begin{aligned}
K_{\text{COMP}}(\mathbf{x}) &= (K_{\text{COMP}} * \delta_x)(\mathbf{x}) \\
&= \int K_{\text{COMP}}(\mathbf{x}')\delta(\mathbf{x} - \mathbf{x}')d\mathbf{x} \\
&\approx \int K_{\text{COMP}}(\mathbf{x}')K_\Phi(\mathbf{x} - \mathbf{x}')d\mathbf{x}
\end{aligned}
\tag{3}
$$

The accuracy of the continuous approximation can be analysed by Taylor series expansion:

$$
\begin{aligned}
K_{\text{COMP}}(\mathbf{x}) &= \int (K_{\text{COMP}}(\mathbf{x}) + \nabla_{\mathbf{x}}K_{\text{COMP}}(\mathbf{x} - \mathbf{x}') \\
&\quad + \frac{1}{2}(\mathbf{x} - \mathbf{x}')\nabla^2 K_{\text{COMP}}(\mathbf{x} - \mathbf{x}') \\
&\quad + \mathcal{O}((\mathbf{x} - \mathbf{x}')^3))K_\Phi(\mathbf{x} - \mathbf{x}')d\mathbf{x} \\
&= K_{\text{COMP}}(\mathbf{x})\int K_\Phi(\mathbf{x} - \mathbf{x}')d\mathbf{x} \\
&\quad + \nabla_{\mathbf{x}}K_{\text{COMP}}(\mathbf{x} - \mathbf{x}')\int (\mathbf{x} - \mathbf{x}')K_\Phi(\mathbf{x} - \mathbf{x}')d\mathbf{x} \\
&\quad + \mathcal{O}((\mathbf{x} - \mathbf{x}')^2)
\end{aligned}
\tag{4}
$$

Ensuring the first-order accuracy of the composing kernel requires the term $\int K_\Phi(\mathbf{x} - \mathbf{x}')d\mathbf{x} = 1$, and the second term in Equation 4 must be 0. This can be accomplished by normalising the feature mapping function and ensuring that it satisfies a symmetry condition. We propose a positive definite Radial Basis Function (RBF) for such a kernel, and its formulation is given by:

$$
\Phi(\mathbf{x}) = \frac{\sum_i^m w_i\varphi(|\mathbf{x} - c_i|)}{\sum_i^m \varphi(|\mathbf{x} - c_i|)}
\tag{5}
$$

where $\mathbf{x} \in R^n$ is the input data, $\mathbf{c} \in R^{n \times m}$ are the centres of the RBFs and are trainable parameters and $w$ is the weight matrix for the feature mapping layer. A natural choice for the RBF can be the Gaussian function, $\varphi(x) = e^{-\frac{|\mathbf{x} - \mathbf{c}|^2}{\sigma^2}}$, where $\sigma$ is a random initialised trainable parameter. If we choose the same number of features as the input size (i.e. $n = m$), this method provides an approximate computation of the desired function value through kernel regression. Unfortunately, in the context of PINNs, the training input size is typically large, making it impractical to scale in this manner. Through empirical study, we demonstrate that a few hundred RBFs prove sufficient to outperform other types of feature mapping functions. During initialisation, $\mathbf{c}$ is sampled from a standard Gaussian distribution.

### 3.1 CONDITIONALLY POSITIVE DEFINITE RBF

In the infinite-width limit, each layer of the Neural Network is treated as a linear system. To guarantee a unique solution, one approach involves introducing conditionally positive definite radial functions by incorporating polynomial terms. The weights serve as Lagrange multipliers, enabling constraints on the RBF coefficients in the parameter space Farazandeh & Mirzaei (2021). We denote this method as RBF-P throughout the paper. Therefore, the feature mapping function is adjusted to:

$$
\Phi(\mathbf{x}) = \frac{\sum_i^m w_i^m\varphi(|\mathbf{x} - c_i|)}{\sum_i^m \varphi(|\mathbf{x} - c_i|)} + \sum_j^k w_j^k P(\mathbf{x})
\tag{6}
$$

Where P is the polynomial function. In the feature mapping layer, it can be represented as:

$$
\begin{bmatrix} f_1 \\ \vdots \\ f_N \end{bmatrix} = \begin{bmatrix} \varphi(r_1^1) & \cdots & \varphi(r_1^m) & | & 1 & x_1 & x^k \\ \vdots & \ddots & \vdots & | & \vdots & \vdots & \vdots \\ \varphi(r_N^1) & \cdots & \varphi(r_N^m) & | & 1 & x_N & x_N^k \end{bmatrix} \begin{bmatrix} W^m \\ - \\ W^k \end{bmatrix}
\tag{7}
$$

where $r = x - c$ and $P$ is the order of the polynomial term.

Based on empirical findings, we observe that the polynomial term is highly effective in nonlinear function approximation, particularly in equations like the Burgers Equation and Navier-Stokes

Equation, all while incurring minimal computational overhead.

By this principle, we can use many other types of RBF without too many restrictions. Other types of RBF are shown in Appendix E Table 5.

## 4 EMPIRICAL RESULTS

Table 1: $L2$ error on varies of PDEs with different feature mapping. The best results are in Blue. Full experimental results with standard deviations are shown in Appendix D.

|           | PINN    | BE      | PE      | FF      | SF      | CT      | CG      | **RBF**  | **RBF-P** |
|-----------|---------|---------|---------|---------|---------|---------|---------|----------|-----------|
| Wave      | 3.73e-1 | 1.04e0  | 1.01e0  | 2.38e-3 | 7.93e-3 | 1.11e0  | 1.04e0  | 2.81e-2  | 2.36e-2   |
| Diffusion | 1.43e-4 | 1.58e-1 | 1.60e-1 | 2.33e-3 | 3.47e-4 | 1.86e0  | 2.72e-2 | 3.07e-4  | 3.50e-5   |
| Heat      | 4.73e-3 | 6.49e-3 | 7.57e-3 | 2.19e-3 | 3.96e-3 | 4.52e-1 | 2.63e-1 | 1.16e-3  | 4.10e-4   |
| Poisson   | 3.62e-3 | 4.96e-1 | 4.91e-1 | 7.59e-4 | 9.08e-4 | 6.35e-1 | 2.33e-1 | 5.26e-4  | 8.94e-4   |
| Burgers   | 1.86e-3 | 5.59e-1 | 5.36e-1 | 7.49e-2 | 1.30e-3 | 9.94e-1 | 7.52e-1 | 2.95e-3  | 3.16e-4   |
| NS        | 5.26e-1 | 7.14e-1 | 6.33e-1 | 6.94e-1 | 3.77e-1 | 5.46e-1 | 4.87e-1 | 2.99e-1  | 2.57e-1   |

Table 2: $L2$ error of the Inverse Problems. * denotes added noises. Full results are in Appendix D.

|            | FF      | SF      | **RBF** | **RBF-P** |
|------------|---------|---------|---------|-----------|
| I-Burgers  | 2.39e-2 | 2.43e-2 | 1.74e-2 | 1.57e-2   |
| I-Lorenz   | 6.51e-3 | 6.39e-3 | 6.08e-3 | 5.99e-3   |
| I-Burgers* | 2.50e-2 | 2.91e-2 | 1.99e-2 | 1.75e-2   |
| I-Lorenz*  | 7.93e-3 | 6.85e-3 | 6.69e-3 | 6.34e-3   |

**Time-dependent PDEs.** Our solution in the Diffusion equation demonstrates superior performance compared to other methods by an order of magnitude. Boundary errors are notably more perceptible in Fourier-based methods, as illustrated in Appendix H, Figure 7.

The RBFs demonstrate enhanced capability in addressing multiscale problems, as illustrated in the Heat equation G.3. In the Heat equation formulation, there exists a substantial contrast in coefficients: $\frac{1}{500\pi^2}$ for the x-direction and $\frac{1}{\pi^2}$ for the y-direction. Figure 8 illustrates that the RBF method effectively preserves the details of the solution at each time step.

**Non-linear PDEs.** We assess the methods using two classic non-linear PDEs: the Burgers equation and the Navier-Stokes equation. Figure 10 illustrates that RBFs with polynomial terms are better in addressing the discontinuity at $x = 0$ in the Burgers equation.

**Inverse Problems.** A major application of the PINNs is able to solve Inverse Problems. The unknown coefficients in the differential equations can be discovered by a small amount of data points. our methods have shown their efficacy in two Inverse Problems, shown in 2

Another experiment aimed to test the robustness of feature mapping functions to noise. $1\%$ Gaussian noises are added to the inverse Burgers problem and $0.5\%$ to the Lorenze system data. The results presented in Table 2 reveal that the four tested feature mapping methods indicate a degree of immunity to noise. Furthermore, RBF-P stands out as the most resilient feature mapping function to noise.

All benchmarked method can be found in Appendix B.

### 4.1 ABLATION STUDY

We carried out ablation studies on the number of RBFs in the feature mapping layer, the number of polynomials for RBF-P and different types of RBFs. Generally, a higher number of RBFs perform better but requires high computation resources. For different cases, the number of polynomials terms required varies. And among all test RBF functions, Gaussians present more stable results. The complete results can be found in Appendix E. The complexity and scalability of different feature mapping functions are included in Appendix F.

## 5  CONCLUSION AND FUTURE WORK

In conclusion, we have introduced a framework for designing an effective feature mapping function in PINNs and proposed Radial Basis Function-based approaches. Our method not only enhances generalisation across a range of forward and inverse physics problems but also surpasses other feature mapping methods by a substantial margin. The RBF feature mapping has the potential to be compatible with various other PINNs techniques, including novel activation functions and loss functions or training strategies such as curriculum training or causal training. While the primary focus of this work has been on solving Partial Differential Equations, the exploration of RBF feature mapping extends to its application in other coordinate-based input neural networks for different tasks.

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

## A  RELATED WORK

**Coordinate Sampling.** As a mesh-free method, PINNs are normally evaluated on scattered collocation points both on the interior domain and IC/BC. Therefore, the sampling strategy is crucial to PINNs' performance and efficiency. A poorly distributed initial sampling can lead to the PDE system being ill-conditioned and NN training instability. The whole design of experiments on the fixed input sampling is reviewed by Das & Tesfamariam (2022). Based on the study of uniform sampling, Wu et al. (2023) proposed an adaptive sampling scheme that refines high residual area during training. Similarly, Importance Sampling inspired by Monte Carlo approximation is investigated by Nabian et al. (2021); Yang et al. (2023). Daw et al. (2022) proposed a novel sampling strategy that mitigates the 'propagation failure' of solutions from IC/BC to the PDE residual field.

**Novel Activation.** The activation function in the MLP has been found to play an important role in the convergence of the PINNs. Popular activation ReLU is deficient for high-order PDEs since its second-order derivative is 0. Apart from the standard Tanh activation Raissi et al. (2019), layerwise and neuron-wise adaptive activation are proven to be useful to accelerate the training Jagtap & Karniadakis (2019); Jagtap et al. (2020). Another line of seminal work, SIREN Sitzmann et al. (2020), which uses periodic activation function, has achieved remarkable results in Neural Representation and tested on solving the Poisson equation. Gaussian Ramasinghe & Lucey (2022) and Gabor Wavelet activations Saragadam et al. (2023) are proven to be effective alternatives.

**Positional Embedding.** Broadly speaking, PINNs can also be considered as a special type of Neural Fields Xie et al. (2021) in visual computing, which specifically feed coordinate-based input to MLPs that represent continuous field quantity (e.g. velocity field in fluid mechanics) over arbitrary spatial and temporal resolution. However, the PINNs community often ignores the fact both perspectives function the same way as Implicit Neural Representations. In the Neural Field, images and 3D shapes are naturally high-frequency signals, whereas deep networks are inherently learning towards the low-frequency components Rahaman et al. (2018). Feature mapping hence has become a standard process in practice that maps the low-dimension coordinates to high-dimension space. The pioneering work was conducted by Rahimi & Recht (2007), who used Fourier features to approximate any stationary kernel principled by Bochner's theorem. the derivative works are done such as Positional Encoding Mildenhall et al. (2020), Random Feature Tancik et al. (2020) and Sinusoidal Feature Sitzmann et al. (2020). Another concurrent work discusses non-periodic feature mapping Zheng et al. (2022); Ramasinghe & Lucey (2021); Wang et al. (2021a). To the best of our knowledge, feature mapping in PINNs has been largely uninvestigated. Only a few work preliminarily adopted Fourier-feature-based methods in PINN Wang et al. (2021b; 2023); Wong et al. (2022).

## B  BENCHMARKED FEATURE MAPPING METHODS

**Basic Encoding:** Mildenhall et al. (2020) $\varphi(x) = [cos(2\pi\sigma^{j/m}x), sin(2\pi\sigma^{j/m}x]^T$ for $j = 0, .., m-1$.

**Positional Encoding:** Mildenhall et al. (2020) $\varphi(x) = [cos(2\pi\sigma^{j/m}x), sin(2\pi\sigma^{j/m}x]^T$ for $j = 0, .., m-1$.

**Random Fourier:** Tancik et al. (2020) $\varphi(x) = [cos(2\pi\sigma\mathcal{B}x), sin(2\pi\sigma\mathcal{B}x)]^T$, where $\mathcal{B} \in \mathbb{R}^{m \times d}$ is sampled from $\mathcal{N}(0,1)$ and $\sigma$ is an arbitrary scaling factor varies case to case.

**Sinusoidal Feature:** Sitzmann et al. (2020) $\varphi(x) = [sin(2\pi\mathbf{W}x + \mathbf{b})]^T$, where $\mathbf{W}$ and $\mathbf{b}$ are trainable parameters.

**Complex Triangle:** Zheng et al. (2022) $\varphi(x) = [max(1 - \frac{|x_1-t|}{0.5d}, 0), max(1 - \frac{|x_2-t|}{0.5d}, 0), \cdots, max(1 - \frac{|x_i-t|}{0.5d}, 0)]^T$, where $t$ is uniformly sampled from 0 to 1.

**Complex Gaussian:** Zheng et al. (2022) $\varphi(x) = [e^{-0.5(x_1-\tau/d)^2/\sigma^2} \bigotimes \cdots \bigotimes e^{-0.5(x_d-\tau/d)^2/\sigma^2}]^T$, where $\tau$ is uniformly sampled from $[0,1]$, and $\bigotimes$ is the Kronecker product.

## C  SPECTRAL BIAS AND COMPOSED NTK

### C.1  SPECTRAL BIAS

Normally PINNs are setup as a standard MLP model $f(\mathbf{x}; \theta)$, and $\theta$ is optimized on the loss function $L(\theta) = \frac{1}{2}|f(\mathbf{x}; \theta) - Y|^2 = \frac{1}{2}\sum_i^N (f(x_i; \theta) - y_i)^2$, where $X$, $Y$ and $\theta$ are training input, training ground truth and model parameters. For an easier formulation, we replace the conventional gradient descent formulation $\theta_{t+1} = \theta_t - \alpha\nabla_\theta L(\theta_t)$ to a gradient flow equation:

$$\frac{d\theta}{dt} = -\alpha\nabla_\theta L(\theta_t) \tag{8}$$

where $\alpha$ should be an infinitesimally small learning rate in the NTK setting.

Given PDE collocation data points $\{x_r^i, \mathcal{D}(\hat{u}_\theta(x_r^i))\}_{i=1}^{N_r}$, and boundary training points $\{x_{bc}^i, \mathcal{B}(\hat{u}_\theta(x_{bc}^i))\}_{i=1}^{N_{bc}}$. The gradient flow can be formulated as Wang et al. (2020):

$$\left[\begin{array}{c} \frac{du(x_b,\theta_t)}{dt} \\ \frac{d\mathcal{L}u(x_r,\theta_t)}{dt} \end{array}\right] = -\left[\begin{array}{cc} \boldsymbol{K}_{uu}^t & \boldsymbol{K}_{ur}^t \\ \boldsymbol{K}_{ru}^t & \boldsymbol{K}_{rr}^t \end{array}\right] \cdot \left[\begin{array}{c} u(x_b,\theta_t) - \mathcal{B}(\hat{u}_\theta(x_{bc})) \\ \mathcal{L}u(x_r,\theta_t) - \mathcal{D}(\hat{u}_\theta(x_r)) \end{array}\right], \tag{9}$$

where the Kernels $\boldsymbol{K}$ are:

$$\begin{aligned} \left(\boldsymbol{K}_{uu}^t\right)_{ij} &= \left\langle \frac{du\left(x_b^i, \theta_t\right)}{d\theta}, \frac{du\left(x_b^j, \theta_t\right)}{d\theta} \right\rangle \\ \left(\boldsymbol{K}_{rr}^t\right)_{ij} &= \left\langle \frac{d\mathcal{L}\left(x_r^i, \theta_t\right)}{d\theta}, \frac{d\mathcal{L}\left(x_r^j, \theta_t\right)}{d\theta} \right\rangle \\ \left(\boldsymbol{K}_{ur}^t\right)_{ij} &= \left(\boldsymbol{K}_{ru}^t\right)_{ij} = \left\langle \frac{du\left(x_b^i, \theta_t\right)}{d\theta}, \frac{d\mathcal{L}u\left(x_r^j, \theta_t\right)}{d\theta} \right\rangle \end{aligned} \tag{10}$$

Since $\boldsymbol{K}$ remains stationary, then $\boldsymbol{K}^t \approx \boldsymbol{K}^0$ as NN width tends to infinity, Equation 9 is rewritten as:

$$\begin{aligned} \left[\begin{array}{c} \frac{du(x_b,\theta_t)}{dt} \\ \frac{d\mathcal{L}u(x_r,\theta_t)}{dt} \end{array}\right] &\approx -\boldsymbol{K}^0 \left[\begin{array}{c} u(x_b,\theta_t) - \mathcal{B}(\hat{u}_\theta(x_{bc})) \\ \mathcal{L}u(x_r,\theta_t) - \mathcal{D}(\hat{u}_\theta(x_r)) \end{array}\right] \\ &\approx (I - e^{-\boldsymbol{K}^0 t}) \cdot \left[\begin{array}{c} \mathcal{B}(\hat{u}_\theta(x_{bc})) \\ \mathcal{D}(\hat{u}_\theta(x_r)) \end{array}\right] \end{aligned} \tag{11}$$

By Schur product theorem, $\boldsymbol{K}^0$ is always Positive Semi-definite, hence it can be Eigen-decomposed to $\boldsymbol{Q}^T\Lambda\boldsymbol{Q}$, where $\boldsymbol{Q}$ is an orthogonal matrix and $\Lambda$ is a diagonal matrix with eigenvalues $\lambda_i$ in the entries. We can rearrange the training error in the form of:

$$\begin{aligned} \left[\begin{array}{c} \frac{du(x_b,\theta_t)}{dt} \\ \frac{d\mathcal{L}u(x_r,\theta_t)}{dt} \end{array}\right] - \left[\begin{array}{c} \mathcal{B}(\hat{u}_\theta(x_{bc})) \\ \mathcal{D}(\hat{u}_\theta(x_r)) \end{array}\right] &\approx (I - e^{-\boldsymbol{K}^0 t}) \cdot \left[\begin{array}{c} \mathcal{B}(\hat{u}_\theta(x_{bc})) \\ \mathcal{D}(\hat{u}_\theta(x_r)) \end{array}\right] - \left[\begin{array}{c} \mathcal{B}(\hat{u}_\theta(x_{bc})) \\ \mathcal{D}(\hat{u}_\theta(x_r)) \end{array}\right] \\ &\approx -\boldsymbol{Q}^T e^{-\Lambda t}\boldsymbol{Q} \cdot \left[\begin{array}{c} \mathcal{B}(\hat{u}_\theta(x_{bc})) \\ \mathcal{D}(\hat{u}_\theta(x_r)) \end{array}\right] \end{aligned} \tag{12}$$

where $e^{-\Lambda t} = \left[\begin{array}{ccc} e^{-\lambda_1 t} & & \\ & \ddots & \\ & & e^{-\lambda_N t} \end{array}\right]$. This indicates the decrease of training error in each component is exponentially proportional to the eigenvalues of the deterministic NTK, and the NN is inherently biased to learn along larger eigenvalues entries of the $\boldsymbol{K}^0$.

## C.2 COMPOSED NTK

The Fourier feature layer is defined as:

$$\varphi(\mathbf{x}) = \left[ a_1 \cos\left(2\pi b_1^{\mathrm{T}}\mathbf{x}\right), a_1 \sin\left(2\pi \mathbf{b}_1^{\mathrm{T}}\mathbf{x}\right), \ldots, a_m \cos\left(2\pi \mathbf{b}_m^{\mathrm{T}}\mathbf{x}\right), a_m \sin\left(2\pi \mathbf{b}_m^{\mathrm{T}}\mathbf{x}\right) \right]^{\mathrm{T}} \quad (13)$$

Hence the NTK is computed by:

$$
\begin{aligned}
\boldsymbol{K}_\Phi\left(x_i, x_j\right) &= \varphi(x_i)^T \varphi(x_j) \\
&= \left[ \begin{array}{c} A_k \cos\left(2\pi \boldsymbol{b_m} x_i\right) \\ A_k \sin\left(2\pi \boldsymbol{b_m} x_j\right) \end{array} \right]^{\mathrm{T}} \cdot \left[ \begin{array}{c} A_k \cos\left(2\pi \boldsymbol{b_m} x_i\right) \\ A_k \sin\left(2\pi \boldsymbol{b_m} x_j\right) \end{array} \right] \\
&= \sum_{k=1}^{m} A_k \cos\left(2\pi b_k^T x_i\right) \cos\left(2\pi b_k^T x_j\right) \\
&\quad + A_k \sin\left(2\pi b_k^T x_i\right) \sin\left(2\pi b_k^T x_j\right) \\
&\boxed{\text{Trigonometric Identities: } \cos(c - d) = \cos c \cos d + \sin c \sin d} \\
&= \sum_{k=1}^{m} A_k^2 \cos\left(2\pi b_k^T \left(x_i - x_j\right)\right).
\end{aligned}
\quad (14)
$$

where $A$ is the Fourier Series coefficients, $\boldsymbol{b}$ is randomly sampled from $\mathcal{N}(0, \sigma^2)$ and $\sigma$ is an arbitrary hyperparameter that controls the bandwidth. Thereafter, the feature space becomes the input of the NTK which gives the identities: $\boldsymbol{K}_{NTK}(x_i^T x_j) = \boldsymbol{K}_{NTK}(\varphi(x_i)^T \varphi(x_j)) = \boldsymbol{K}_{NTK}(\boldsymbol{K}_\Phi(x_i - x_j))$.

# D  COMPLETE EXPERIMENTAL RESULTS FOR TABLE 1&2

## D.1  COMPLETE RESULTS FOR TABLE 1

Table 3: Full PDEs benchmark results comparing different feature mapping methods in $L2$ error. The best results are in  Blue . Standard deviations are shown after $\pm$.

|           | PINN             | BE               | PE               | FF               | SF               |
|-----------|------------------|------------------|------------------|------------------|------------------|
| Wave      | 3.73e-1±2.37e-2  | 1.04e0±3.55e-1   | 1.01e0±4.02e-1   | 2.38e-3±3.75e-4  | 7.93e-3±9.32e-4  |
| Diffusion | 1.43e-4±4.84e-5  | 1.58e-1±6.13e-2  | 1.60e-1±1.20e-2  | 2.33e-3±7.51e-4  | 3.47e-4±6.11e-5  |
| Heat      | 4.73e-3±6.14e-5  | 6.49e-3±6.37e-4  | 7.57e-3±1.02e-4  | 2.19e-3±3.12e-4  | 3.96e-3±2.56e-4  |
| Poisson   | 3.62e-3±1.24e-4  | 4.96e-1±2.15e-2  | 4.91e-1±1.08e-2  | 7.58e-4±9.01e-5  | 9.07e-4±1.02e-5  |
| Burgers   | 1.86e-3±1.20e-4  | 5.58e-1±2.57e-2  | 5.36e-1±3.70e-2  | 7.50e-2±5.15e-3  | 1.30e-3±6.21e-4  |
| Steady NS | 5.26e-1±1.01e-2  | 7.14e-1±1.33e-2  | 6.33e-1±2.35e-2  | 6.94e-1±1.06e-3  | 3.77e-1±2.37e-2  |

|           | CT               | CG               | RBF              | RBF-P            |
|-----------|------------------|------------------|------------------|------------------|
| Wave      | 1.11e0±3.21e-2   | 1.03e0±1.05e-2   | 2.81e-2±3.67e-3  | 2.36e-2±1.59e-2  |
| Diffusion | 1.86e0±2.31e-2   | 2.72e-2±1.02e-1  | 3.06e-4±9.51e-6  | 3.49e-5±6.54e-6  |
| Heat      | 4.52e-1±6.51e-2  | 2.62e-1±2.36e-2  | 1.15e-3±1.02e-4  | 4.09e-4±9.62e-6  |
| Poisson   | 6.34e-1±3.04e-1  | 2.33e-1±5.47e-2  | 5.25e-4±6.24e-5  | 8.94e-4± 6.51e-5 |
| Burgers   | 9.93e-1±4.51e-2  | 7.52e-1±3.24e-2  | 2.94e-3±2.35e-4  | 3.15e-4±2.14e-5  |
| Steady NS | 5.46e-1±2.35e-2  | 4.86e-1±3.65e-2  | 2.99e-1±6.51e-2  | 2.56e-1±6.21e-2  |

## D.2  COMPLETE RESULTS FOR TABLE 2

Table 4: Full Benchmark results on the Inverse problems in $L2$ error.  * indicates problems with noises added to the data.

|            | FF               | SF               | RBF              | RBF-P            |
|------------|------------------|------------------|------------------|------------------|
| I-Burgers  | 2.39e-2±9.64e-4  | 2.43e-2±4.678e-3 | 1.74e-2±6.57e-3  | 1.57e-2±9.36e-4  |
| I-Lorenz   | 6.51e-3±7.65e-4  | 6.39e-3±6.21e-4  | 6.08e-3±3.69e-4  | 5.99e-3±2.31e-4  |
| I-Burgers* | 2.50e-2±6.32e-3  | 2.91e-2±2.69e-3  | 1.99e-2±3.62e-3  | 1.75e-2±5.63e-3  |
| I-Lorenz*  | 7.93e-3±8.65e-4  | 6.85e-3±6.36e-4  | 6.69e-3±5.20e-4  | 6.34e-3±8.61e-4  |

# E  ABLATION STUDY

In this section, we show some additional experiments on our RBF feature mapping including investigations on the Number of RBFs, Number of Polynomials and different RBF types.

## E.1  NUMBER OF RBFS

Figure 2 has shown generally more RBFs (256) yield better results. It however does demand a higher memory and can be slow in some cases. It shows in the Diffusion equation, with 256 RBFs, the error reduces quite significantly. Otherwise, it only has limited improvements because the error is already very low. We use 128 RBFs in general case for a better performance-speed tradeoff.

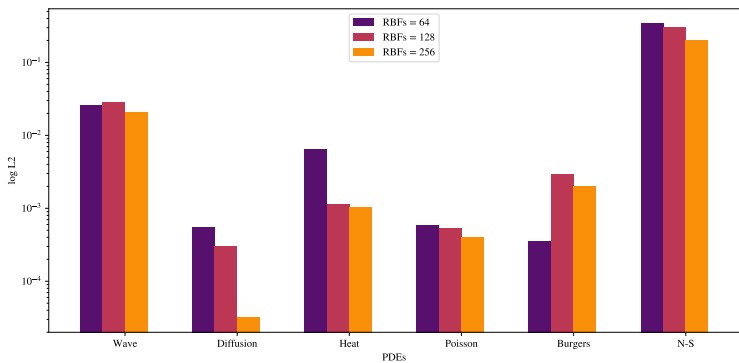

Figure 2: Ablation study on different number of RBFs

## E.2  NUMBER OF POLYNOMIALS

Figure 3 shows an ablation study of how the number of polynomials in feature mappings influences performance in PDEs. It has shown RBF feature mapping with 20 polynomials has achieved best results in the Diffusion equation, Poisson equation and N-S equation. And 10 polynomial terms are better in Heat equation and Burgers equation, thought its performance is matching with only 5 polynomials.

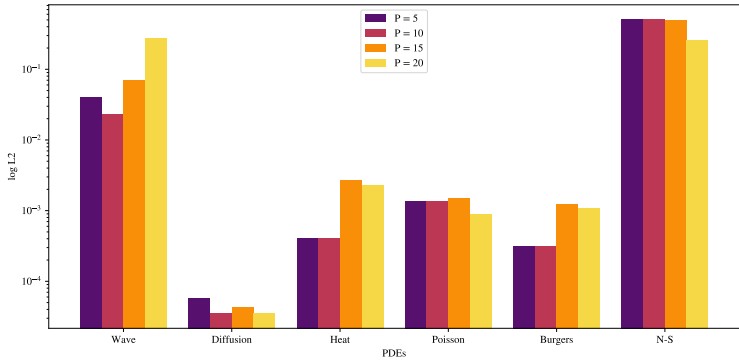

Figure 3: Ablation study on different number of polynomials

### E.3 Different Types of RBFs

Following Table 5 are common positive definite Radial Basis Functions.

Table 5: Types of Radial Basis function and their formulation. $\mathbf{x} - \mathbf{c}$ is shorten as r.

| Type | Radial function |
|------|-----------------|
| Cubic | $r^3$ |
| TPS(Thin Plate Spline) | $r^2 log(r)$ |
| GA(Gaussian) | $e^{-r^2/\sigma^2}$ |
| MQ(Multiquadric) | $\sqrt{1 + r^2}$ |
| IMQ(Inverse MQ) | $1/\sqrt{1 + r^2}$ |

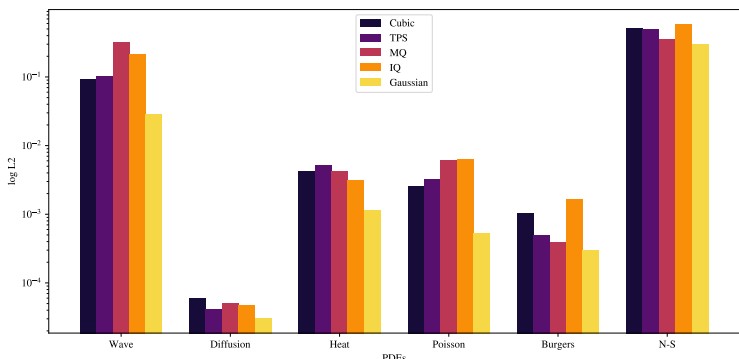

Figure 4: Ablation study on different types of RBFs

The Figure 4 has shown Gaussian RBF is dominating all types of PDEs. However other types of RBF are in similar performance. We generally prefer Gaussian RBF in all cases due to its nice properties.

## F Complexity and Scalability analysis

The comparison of complexity and scalability of feature mapping methods are shown in this section.

Although all feature mapping methods are similar in computational complexity, for completeness, we include the complexity of the feature layers that map 128 features and 4 fully connected layers with 50 neurons each.

Table 6: Computational complexity

|        | FF    | SF    | RBF-I | RBF-P-5 | RBF-P-10 | RBF-P-15 | RBF-P-20 |
|--------|-------|-------|-------|---------|----------|----------|----------|
| FLOPs  | 139.5M | 142.1M | 139.5M | 142.5M | 145.0M | 147.5M | 150.0M |
| Params | 14.2k | 14.3k | 14.2k | 14.5k | 14.7k | 14.9k | 15.2k |

Due to software optimisation and package compatibility, the feature mapping methods can have very different computational efficiency in training. To demonstrate, we run the above models on different numbers of sample points on Diffusion equation for 3 times in different random seeds.RBF-COM stands for compact support RBF, and RBF-P uses 20 polynomials.
The time consumed by Fourier Features is noticeably higher than other methods. All methods have similar runtime for sample points less than $1e4$, that is because all sample points computed are within a GPU parallelisation capacity.

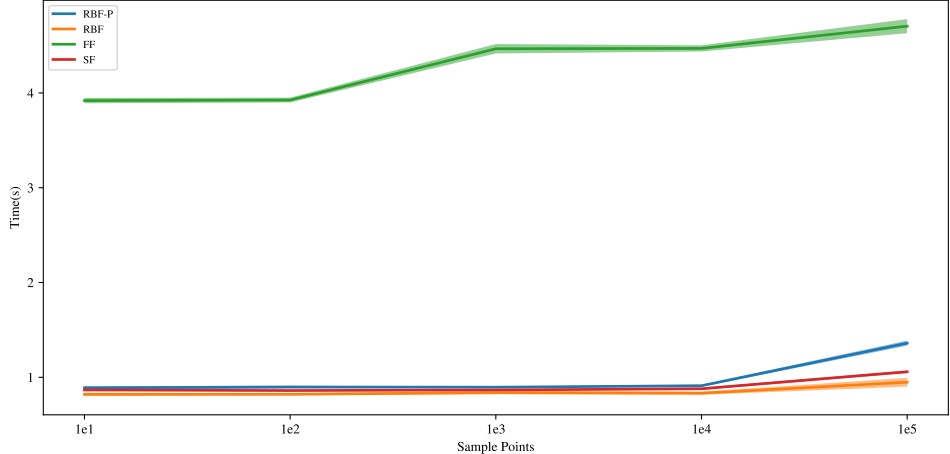

Figure 5: Time consumption on different numbers of sample points with different feature mapping methods

## G    BENCHMARK PDEs AND BOUNDARY CONDITIONS

### G.1    WAVE EQUATION

The one-dimensional Wave Equation is given by:
$$u_{tt} - 4u_{xx} = 0. \tag{15}$$

In the domain of:
$$(x, t) \in \Omega \times T = [-1, 1] \times [0, 1]. \tag{16}$$

Boundary condition:
$$u(0, t) = u(1, t) = 0. \tag{17}$$

Initial condition:
$$u(x, 0) = \sin(\pi x) + \frac{1}{2}\sin(4\pi x) \tag{18}$$
$$u_t = 0 \tag{19}$$
$$\tag{20}$$

The analytical solution of the equation is:
$$u(x, t) = \sin(\pi x)cos(2\pi t) + \frac{1}{2}\sin(4\pi x)cos(8\pi t). \tag{21}$$

### G.2    DIFFUSION EQUATION

The one-dimensional Diffusion Equation is given by:
$$u_t - u_{xx} + e^{-t}(sin(\pi x) + \pi^2 sin(\pi x)) = 0 \tag{22}$$

In the domain of:
$$(x, t) \in \Omega \times T = [-1, 1] \times [0, 1]. \tag{23}$$

Boundary condition:
$$u(-1, t) = u(1, t) = 0 \tag{24}$$

Initial condition:
$$u(x, 0) = sin(\pi x) \tag{25}$$

The analytical solution of the equation is:
$$u(x, t) = e^t sin(\pi x) \tag{26}$$

where $\alpha = 0.4, L = 1, n = 1$

### G.3   HEAT EQUATION

The two-dimensional Heat Equation is given by:

$$u_t - \frac{1}{(500\pi)^2} u_{xx} - \frac{1}{\pi^2} u_{yy} = 0. \tag{27}$$

In the domain of:

$$(\mathbf{x}, t) \in \Omega \times T = [0, 1]^2 \times [0, 5]. \tag{28}$$

Boundary condition:

$$u(x, y, t) = 0. \tag{29}$$

Initial condition:

$$u(x, y, 0) = \sin(20\pi x) \sin(\pi y). \tag{30}$$

### G.4   POISSON EQUATION

The two-dimensional Poisson Equation is given by:

$$-\Delta u = 0 \tag{31}$$

In the domain of:

$$\mathbf{x} \in \Omega = \Omega_{rec} \backslash R_i. \tag{32}$$

where

$$\Omega_{rec} = [-0.5, 0.5]^2, \tag{33}$$
$$R_1 = [(x, y) : (x - 0.3)^2 + (y - 0.3)^2 \le 0.1^2], \tag{34}$$
$$R_2 = [(x, y) : (x + 0.3)^2 + (y - 0.3)^2 \le 0.1^2], \tag{35}$$
$$R_3 = [(x, y) : (x - 0.3)^2 + (y + 0.3)^2 \le 0.1^2], \tag{36}$$
$$R_4 = [(x, y) : (x + 0.3)^2 + (y + 0.3)^2 \le 0.1^2]. \tag{37}$$

Boundary condition:

$$u = 0, x \in \partial R_i, \tag{38}$$
$$u = 1, x \in \partial \Omega_{rec}. \tag{39}$$

### G.5   BURGERS EQUATION

The one-dimensional Burgers Equation is given by:

$$u_t + u u_x = \nu u_{xx} \tag{40}$$

In the domain of:

$$(x, t) \in \Omega = [-1, 1] \times [0, 1]. \tag{41}$$

Boundary condition:

$$u(-1, t) = u(1, t) = 0. \tag{42}$$

Initial condition:

$$u(x, 0) = -\sin \pi x \tag{43}$$

where $\nu = \frac{0.01}{\pi}$

## G.6 STEADY NS

The steady incompressible Navier Stokes Equation is given by:
$$\nabla \cdot \mathbf{u} = 0, \tag{44}$$
$$\mathbf{u} \cdot \nabla \mathbf{u} + \nabla p - \frac{1}{\text{Re}} \Delta \mathbf{u} = 0. \tag{45}$$
$$\tag{46}$$

In the domain(back step flow) of:
$$\mathbf{x} \in \Omega = [0,4] \times [0,2] \setminus ([0,2] \times [1,2] \cup R_i) \tag{47}$$
Boundary condition:
$$\text{no-slip condition:} \quad \mathbf{u} = 0. \tag{48}$$
$$\text{inlet:} \quad u_x = 4y(1-y), u_y = 0. \tag{49}$$
$$\text{outlet:} \quad p = 0. \tag{50}$$
where $Re = 100$

## G.7 nD POISSON EQUATION

The nth-dimensional Poisson Equation is given by:
$$-\Delta u = \frac{\pi^2}{4} \sum_{i=1}^{n} \sin\left(\frac{\pi}{2} x_i\right) \tag{51}$$

In the domain of:
$$x \in \Omega = [0,1]^n \tag{52}$$
Boundary condition:
$$u = 0 \tag{53}$$
The analytical solution of the equation is:
$$u = \sum_{i=1}^{n} \sin\left(\frac{\pi}{2} x_i\right) \tag{54}$$

## G.8 INVERSE BURGERS EQUATION

The one-dimensional Inverse Burgers Equation is given by:
$$u_t + \mu_1 u u_x = \mu_2 u_{xx} \tag{55}$$
In the domain of:
$$(x,t) \in \Omega = [-1,1] \times [0,1]. \tag{56}$$
Boundary condition:
$$u(-1,t) = u(1,t) = 0. \tag{57}$$
Initial condition:
$$u(x,0) = -\sin \pi x \tag{58}$$
where $\mu_1 = 1$ and $\mu_2 = \frac{0.01}{\pi}$

## G.9 INVERSE LORENZ EQUATION

The 1st-order three-dimensional Lorenz Equation is given by:
$$\begin{aligned}
\frac{dx}{dt} &= \alpha(y-x), \\
\frac{dy}{dt} &= x(\rho - z) - y, \\
\frac{dz}{dt} &= xy - \beta z,
\end{aligned} \tag{59}$$

where $\alpha = 10$, $\beta = \frac{8}{3}$, $\rho = 15$ and the initial points are $x_0 = 0, y_0 = 1, z_0 = 1.05$.

## H VISUALISATIONS OF PDES SOLUTION

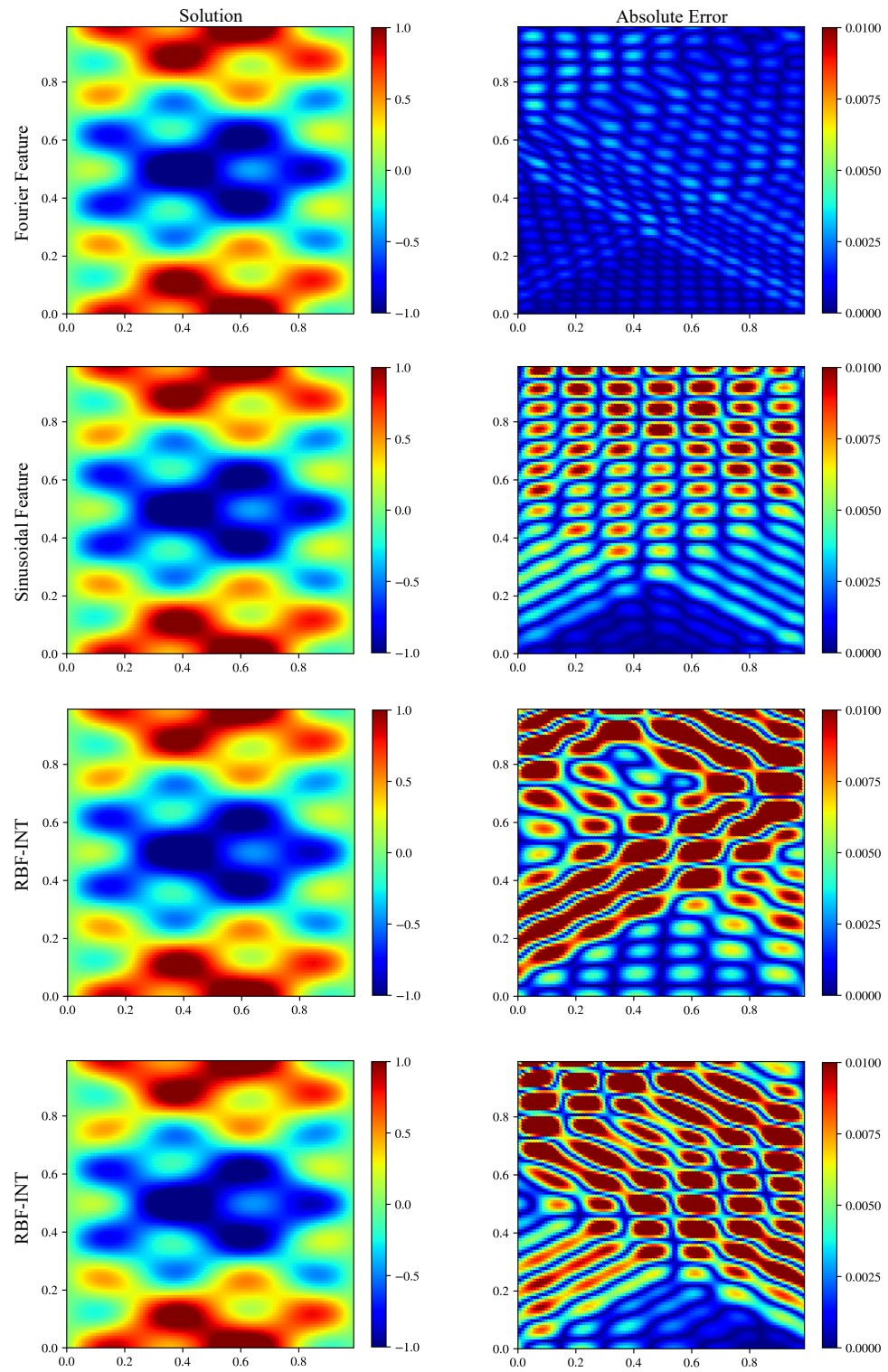

Figure 6: Wave equation

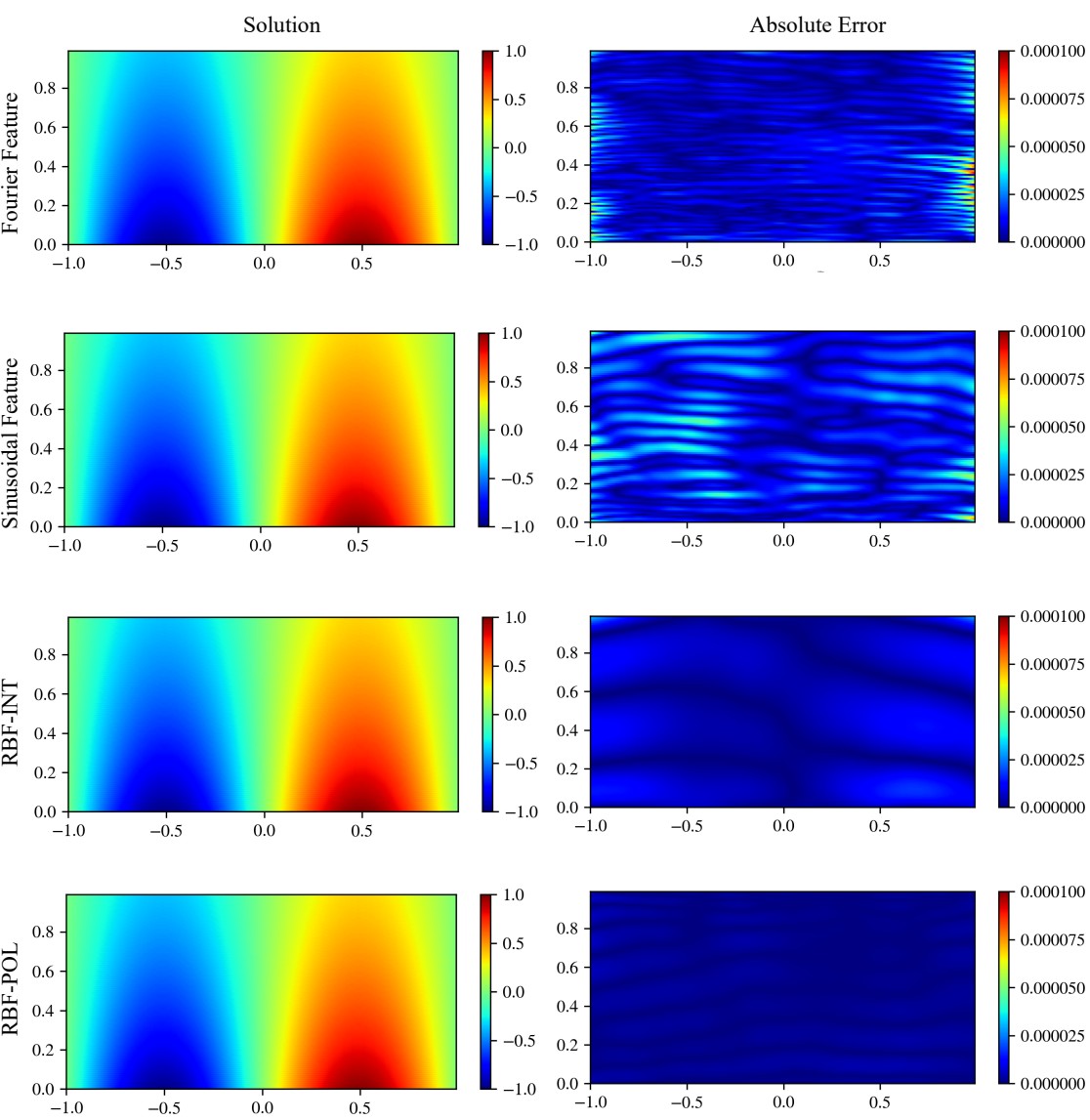

Figure 7: Diffusion equation

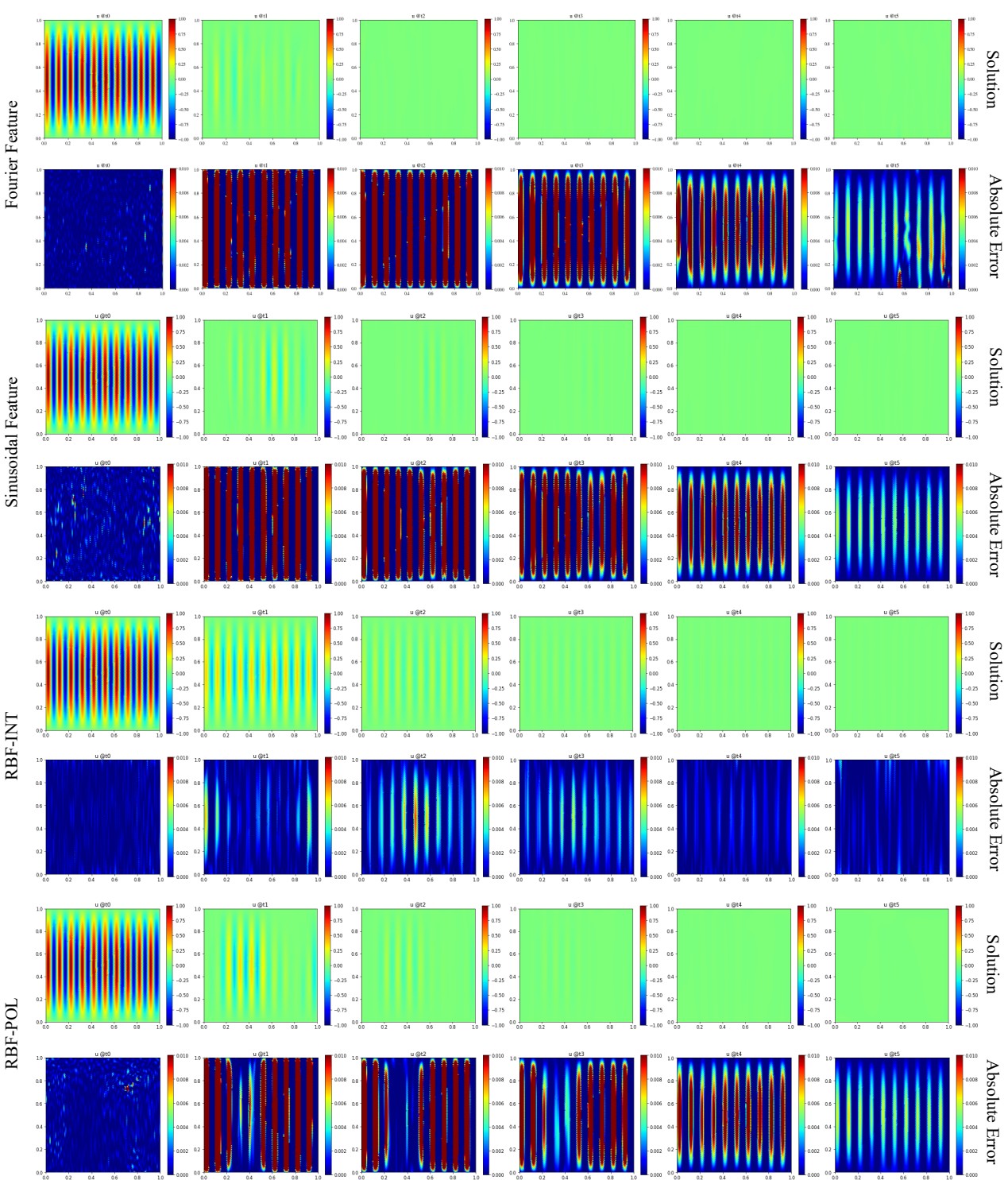

Figure 8: Heat equation

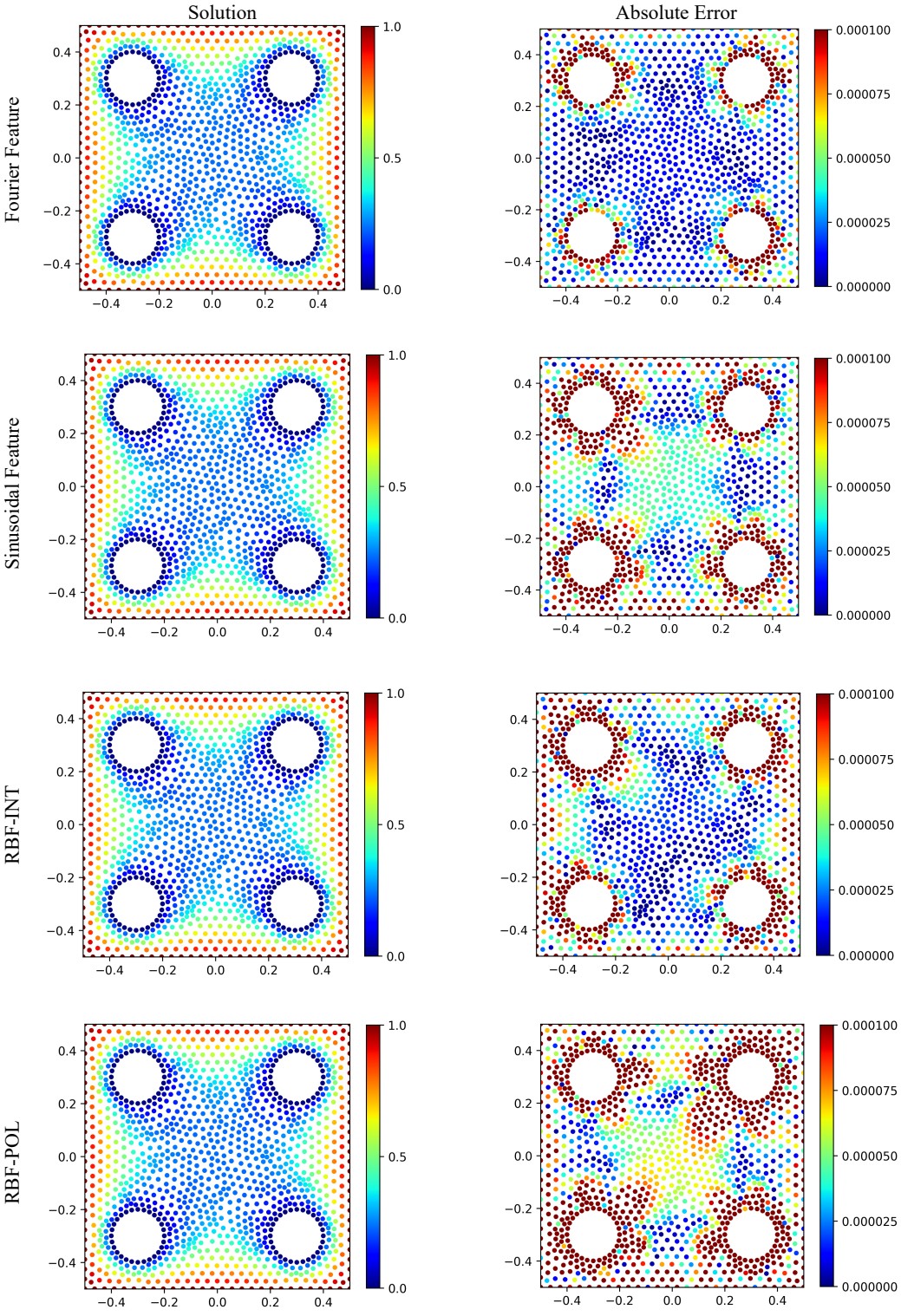

Figure 9: Poisson equation

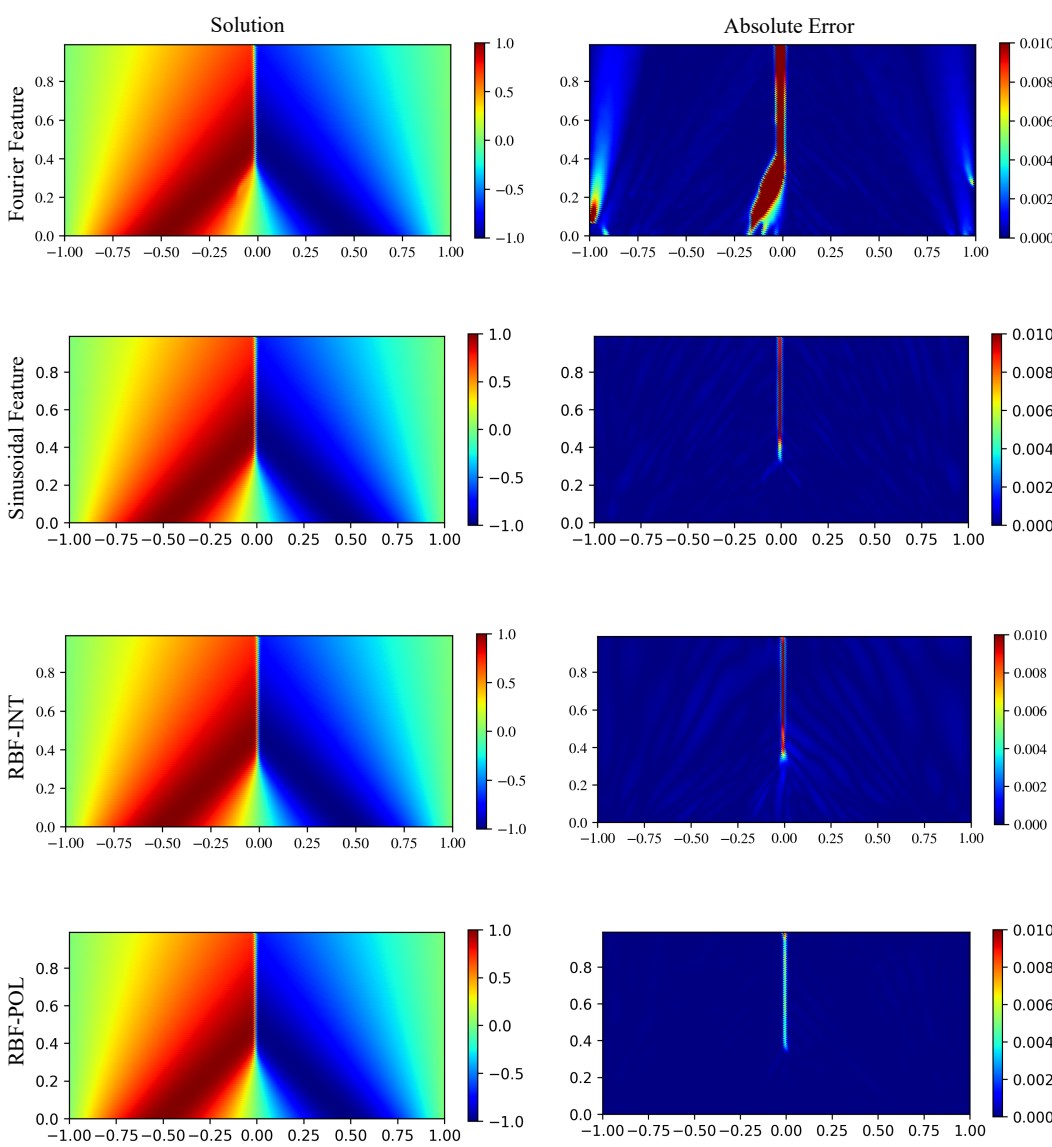

Figure 10: Burgers equation

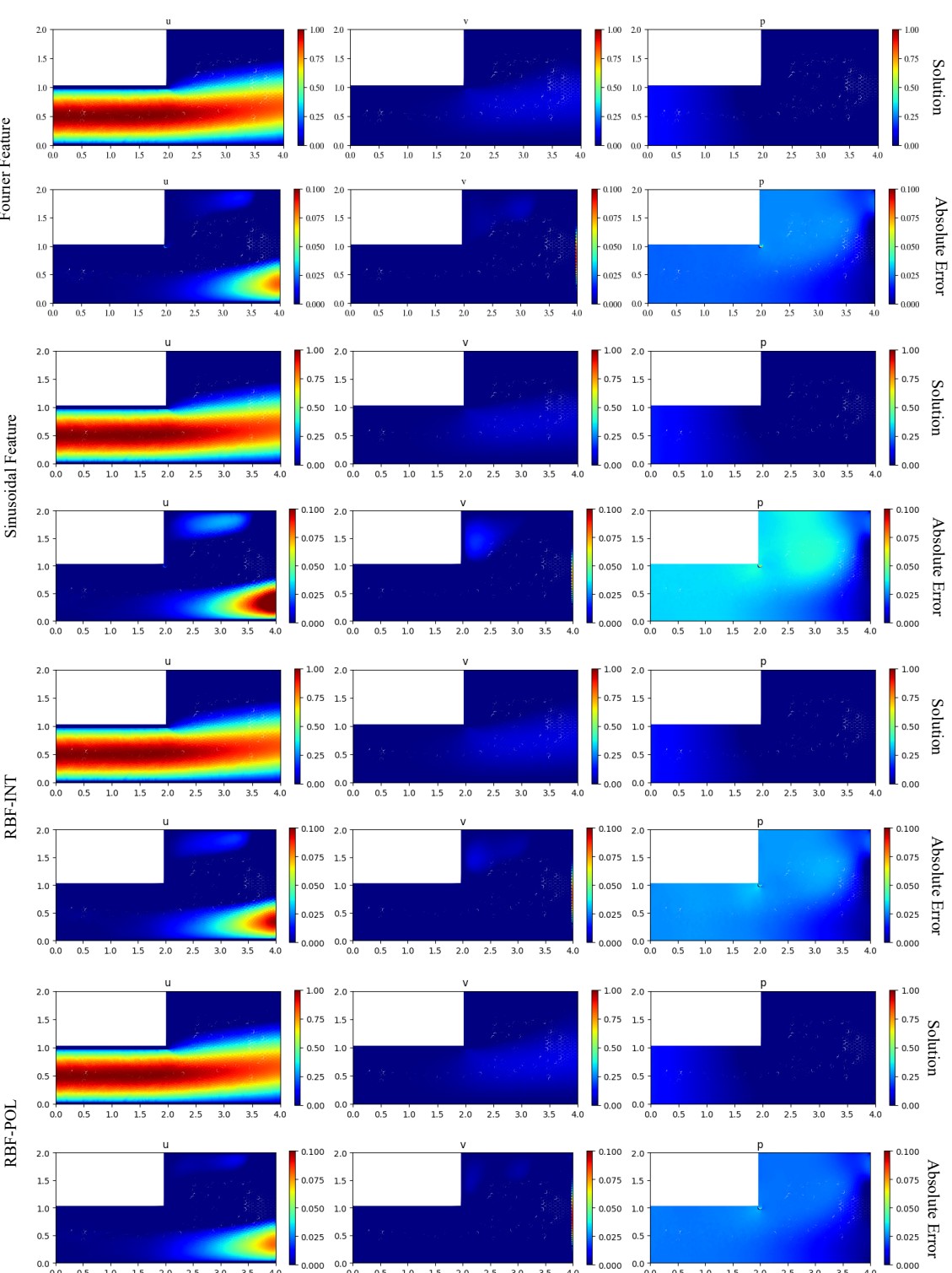

Figure 11: Navier-Stokes equation

