# OpenReview forum: "RBF-PINN: NON-FOURIER POSITIONAL EMBEDDING IN PHYSICS-INFORMED NEURAL NETWORKS"
_ICLR.cc/2024/Workshop/AI4DiffEqtnsInSci — AI4DiffEqtnsInSci @ ICLR 2024 Poster_

### Official Review · Reviewer_j7ty · 2024-02-19
**Feature Mapping Scheme for PINNs**

**Rating:** 6
**Confidence:** 3

**Review:**

The authors propose a feature mapping scheme termed RBF-PINN that uses radial basis functions to map/transform the inputs of a PINN network. They apply this to six cases for solving the forwards problem and four cases for solving the inverse problem. Their approach is contrasted with the original PINNs approach and several other feature mapping approaches.

I believe the authors have done a lot of work and the paper for the most part is clear to the reader.

I am familiar with most of the discussed concepts in the paper (e.g. PINNs and feature mapping) but I have followed only to some extend the literature on PINNs to make strong claims regarding the novelty of the work. One of the things I would expect is to have a comparison with the work ``Adaptive activation functions accelerate convergence in deep and physics-informed neural networks’’ but since it is mentioned that this future step I don’t think is necessary here.

My main comments for the paper are:

(i)	Even though the paper is well written there are some cases the notation becomes confusing in my opinion. The author start by using the usual notation u(x,t) for the (time-dependent) PDEs and in equation (2) x_r for the collocation points. However, when they introduce Φ it seems to depend only on x (or bold x). This in my opinion might create some confusion.
(ii)	Equation 5, a definition of $w_i$ is missing.
(iii)	Regarding the conditionally positive definite RBF. The authors introduce polynomial terms, and they claim this guarantees uniqueness. I would like to see a few references there that support this claim.
(iv)	Can the authors comment if there cases for which the introduction of the polynomial terms can lead to “explosions” of the gradient or to vanishing gradients? For example, when the degree of polynomial is large.
(v)	How the authors chose the parameter σ in the kernel?
(vi)	In a few places in the text the authors mention the $l2$ loss where the corresponding figures shows $L_2$. Make things consistent. (There are figures also in the Appendix with this inconsistency).
(vii)	Figure 19 it’s not clearly readable. Perhaps the authors can make some changes there.
(viii)	Section Limitation of Fourier Features “Navier-Stokes” is misspelled.
(ix)	Perhaps, for future work, it might be of interest to the authors instead of sampling the coefficients c randomly to use approaches such the “Sampling weights of deep neural networks)”.

---

### Official Review · Reviewer_tfx5 · 2024-02-24

**Rating:** 5
**Confidence:** 4

**Review:**

This paper investigates the benefits of feature mapping in physics-informed neural networks. The authors show that Fourier-based feature mapping has certain limitations, and conditionally positive definite radial basis functions provide better results for solving partial differential equations using PINNs. Different PDEs are investigated in this work for both forward and inverse problems, and overall results with RBF-based feature mapping are better compared to other methods. However, the motivation behind using feature mapping in PINNs and how it exactly addresses the limitations of PINNs are missing. Addressing this would enhance the credibility of the proposed approach. There are several critical points that require attention to strengthen the paper's contribution and clarity.

1. The error for PINNs and the proposed RBF is similar for even and uneven sampling for nD Poisson equations. What is the justification for using RBF for feature mapping for this problem?
2. MLP on page 2 is not defined.
3. There is no visual comparison of solutions from PINNs in Figs 6-11.
4. The number of RBFs does not seem to have a significant impact on L2 error, and in the case of Burger’s equation, the error seems to increase as the number of RBFs is increased. Is it due to overfitting? Similar observations for the number of polynomials are also noted.
5. In the conclusion, the authors mentioned that this paper introduces a framework. However, the present paper does not discuss how to decide the feature mapping depending on the complexity of the problem. Hence, we cannot call the present study on feature mapping a framework.
6. In the conclusion section, the author mentioned that the RBF feature mapping enhances generalization across forward and inverse problems. However, the paper does not discuss anything about the errors for out-of-distribution data.
7. Detailed experimental settings, specific implementations for each numerical experiment, and hyperparameters are not provided. The code is also not made open-source, making it difficult to reproduce the results.

---

### Meta-Review · Area_Chair_3WaL · 2024-03-03

**Recommendation:** Accept (Poster)

**Metareview:**

This paper is marked as borderline. I vote for acceptance as poster pending that the authors address the reviewers' comments in the camera-ready verion.

---

### Decision · Program_Chairs · 2024-03-03

Accept (Poster)